# Earth-Based Building Incorporating *Sargassum muticum* Seaweed: Mechanical and Hygrothermal Performances

Houssam Affan [1], Karim Touati [1,2], Mohammed-Hichem Benzaama [3], Daniel Chateigner [4] and Yassine El Mendili [5,*]

1 Builders Lab, COMUE Normandie Université, Builders Ecole d'Ingénieurs, 1 Rue Pierre et Marie Curie, 14610 Epron, France
2 EPF Ecole d'Ingénieurs, 21 Boulevard Berthelot, 34000 Montpellier, France
3 CEREMA Normandie Centre, 10 Chemin de la Poudrière, 76120 Le Grand-Quevilly, France
4 CRISMAT, CNRS UMR 6508, ENSICAEN, IUT Caen, Université de Caen Normandie, Normandie Université, 6 Bd Maréchal Juin, CEDEX 4, 14050 Caen, France
5 Institut de Recherche en Constructibilité IRC, Ecole Spéciale des Travaux Publics, 28 Avenue du Président Wilson, CEDEX, 94234 Cachan, France
* Correspondence: yelmendili@estp-paris.eu; Tel.: +33-1-49-08-03-20

**Abstract:** Once the tide recedes and leaves a significant amount of stranded seaweed on the coast, marine macroalgae pose a serious threat to the surrounding area. Through this work, we considered a large-scale application of stranded macroalgae in building construction. For the first time we studied the impact of incorporating *Sargassum mitucum* seaweed fiber in replacement of flax fiber used for a standard structural cob. Thus, cob specimens were elaborated and analyzed to evaluate their compressive and hygrothermal performances. It was found that the compressive strength and water vapor resistance factors of cob decreased with the algae content. Additionally, the obtained results showed that a cob made with *Sargassum muticum* algae presented better thermal (insulation and inertia) and hygroscopic properties than those of a cob made with a flax fiber. Indeed, the replacement of flax straw by algae lead to a reduction in the thermal conductivity by 38% when compared to the standard cob with 2.5% of flax straw fiber. Consequently, numerical simulation showed a reduction in the energy needs in buildings made with an algae-based cob when compared to those made with a flax-based cob. This study can contribute to a global environmental and economic issue, i.e., the valorization of brown algae on a large scale. Indeed, the worldwide knows the largest sea of sargassum algae extent measures over 8850 km$^2$. This huge mass of brownish algae is expanding every year, which now covers an area from Africa to the Caribbean. It weighs more than 20 million tons and extends from the Gulf of Mexico to the west coast of Africa. We show that stranded algae, which are considered as wastes, have the ability to improve the mechanical and hygrothermal performance of cob-based material.

**Keywords:** earth construction; *Sargassum muticum*; algae fiber; mechanical and hygrothermal performance; energy needs

## 1. Introduction

In order to reduce $CO_2$ emissions in the construction industry, a number of cementing system alternatives have been proposed, including (a) switching from coal to natural gas in the Portland cement preparation process, which is based on a chemical absorption process capable of capturing $CO_2$, (b) substituting cement with supplementary cementitious materials, and (c) replacing cement with natural materials, such as earth [1–5].

The use of earth as building material predates ten thousand years, and numerous historic earthen constructions, such as the Great Wall of China, the Egyptian Pyramids, and the monumental works of Iran and Iraq, are still standing today [6,7]. Earthen construction refers to the practice of using earth as the primary building material. It can

be seen as a stand-alone natural building concept or as a collection of technologies that enhance conventional modern construction techniques with high temperature retention, durability, environmental sustainability, and more "methodically" [8–10]. Individuals and communities all over the world are using earth building to decrease their contributions to climate change and global warming [11]. Some earth-building techniques that make use of soil as a building material include adobe or mud brick, rammed earth, cob, poured earth, and pressed earth [12,13]. This type of construction has many interesting features, mainly its capacity to stabilize humidity in the building and to present a high thermal mass, as well as its very low economic and ecological cost. As a primary material, with no additives, the earth is an excellent controller of humidity because of its capacity of absorption and desorption. It captures or releases air moisture according to the ambient humidity, thus favoring a healthy indoor environment [14–17].

Cob is the most popular earth-building technique in Normandy (France), which combines clay, sand, water, and fibers [9]. For cob construction, straw and stones are mixed into sandy soil [9]. The fiber content of a fresh cob is typically between 20 and 30 kg/m$^3$, with fiber lengths ranging from 300 to 500 mm [17]. The sandy soil is mixed with water until it becomes plastic, and the straw fibers are then acted underfoot by livestock hooves. The cob is then stacked and dried into 700–800 mm high walls. The wall faces are cut vertically with a spade once the masses have the appropriate moisture content [17]. Typically, the traditional cob has a density of 1200 to 1700 kg/m$^3$ [9,18]. Traditional cob making is sustainable, economical, and friendly to the environment because it employs locally available soils and fibers and needs very little water and energy.

The structural behavior of earth buildings can be influenced by a variety of environmental conditions. An increased water content weakens a material's strength. Additionally, the clayey soil that is usually used for earth building exhibits differential settlements, a weak shear strength, and high compressibility, all of which must be stabilized to increase its mechanical performance [19]. Due to its relatively low strength and substantial plastic shrinkage rate, the earth has a high risk of cracking. A successful method to stabilize the earth and to regulate fracture initiation and propagation in earth caused by plastic and autogenous shrinkage is the inclusion of fibers with arbitrary distribution [20,21]. By supplying bridging forces across the cracks, adding fibers to earth improves its ability to reduce strain, reduce aggregate segregation, and enhance the compressive strength [20,21]. In order to improve the efficiency of earth concrete, natural fibers have been frequently used in earth construction.

Despite the fact that unfibered cob has been reported [20–22], the cob technique was typically linked to the addition of a natural fiber. The most frequently mentioned fiber used for cob is flax straw. According to Petitjean [23], changes in agricultural methods during the 19th century led to an overabundance of straw, which encouraged its usage at the expense of earlier-used fibers. In reality, a wide variety of fibers were utilized to build traditional cobs. Since the fibers were gathered locally [24], this variability illustrates how past builders customized the vernacular cob construction method to take advantage of the materials that were readily available in their immediate surroundings.

Algae fibers are widely available and are not recycled, unlike flax fibers and the sunflower plant [25,26]. They are completely renewable and biodegradable, and as a result, they can serve as a great starting point for the development of resource-saving alternatives in the building sector. They have the potential to drastically lower buildings' carbon footprints, particularly when used in conjunction with effective earth structures.

Seaweeds are sustainable materials that are not only available in many parts of the world, but also can be collected easily on the seaside. Seaweed has many advantages: it is non-toxic and fireproof, it offers good insulation, it reduces $CO_2$ emissions, it has a life expectancy of over 150 years, it is valuable, and it can be optically appealing [27].

A "green tide" phenomenon has appeared on the coasts of Bretagne and western France since the end of the twentieth century [28]. A green tide is a proliferation of green algae that can cover the coastline due to a high concentration of specimens. The seaweed

in question is a member of the genus Ulva. Natural conditions such as luminosity, high water, and high temperature are favorable for the development of green, brown, and red algae from May to September [29]. In addition, this tidal zone has a deteriorate effect on building materials [1–4]. The main cause of the seaweed proliferation, however, remains the abundant contribution of nitrates from rivers to the ocean [30] of agricultural origin. These nitrates are nutritive elements for algae and they greatly enhance their growth. Stranded seaweeds and the rafts formed by algal remaining on the water's surface are a common problem for Normandy shellfish farming, but they also endanger naval activity [31]. The algae are hazardous during decomposition because they form a crust that emits gases [31]. They produce the toxic $H_2S$ gas with a strong odor by anaerobic fermenting, which has caused several human and animal deaths in Brittany. As the seaweeds grows in number, the health and environmental consequences become more severe. To address this issue, communities collect stranded algae or push it back into the sea [29].

Indeed, according to the Centre d'Etudes et de Valorisation des Algues (CEVA), coastal localities from the Ile de Ré to Basse Normandie have noticed an increase in mudflats on the coast as well as more and more frequent strandings. Between 50,000 and 100,000 $m^3$ of algae are collected and treated annually by the communities concerned [29]. This activity has generated a total cost of about EUR 1.7 million per year on the coastline from Normandy to Ile de Ré alone (cost of 20 EURper $m^3$) [29]. Research based on invasive algae such as Sargassum muticum that were conducted on a small geographic region has demonstrated that this species can proliferate in extremely significant amounts, especially in the summer, when stocks can reach 2200 tons in July on only 3 $km^2$ of surface [29]. Thus, it can be estimated that these quantities are largely sufficient to be exploited on a large scale [32], and even more so if the methods proposed here are continued on other species of algae (green algae for example). Among others, the authors, in the framework of a 3-year project named SNOTRA and co-financed by the Normandy region, have estimated average annual volumes of 12,000 tons of fresh *Sargassum muticum* [33] or 1200 tons of dry matter. As an extension of SNOTRA, the SAVENT project, which is co-funded by ADEME and conducted by ALGAIA SA, has also studied the potential use of seaweed fibers for insulation. These *Sargassum muticum* quantities are significant because they account for one-sixth of the total volume of seaweed collected in France each year. At the moment, the only approach to address this issue is via the removal of stranded seaweed and rafts. Other ways of recovery such as the collection of seaweed by the agricultural sector for the purpose of amendments, plant nutrients, composts, or cosmetics [32] are possible, but not yet fully operational.

On a global scale, the world is witnessing the largest sargassum seaweed sea, which spans over 8850 $km^2$ [34]. It has been called the "Great Atlantic Sargassum Belt" by scientists. This massive swarm of brownish seaweed is growing by the year and now expands from Africa to the Caribbean. It weighs more than 20 million tons and expands from the Gulf of Mexico to Africa's west coast.

The properties of seaweed have sparked interest to incorporate these materials into the field of construction. Red seaweed is used in the construction industry in the form of powder and gel. On the other side, brown seaweeds are used in the form of fibers, powder, and alginates. Red seaweed that is used in a powder or gel form in a small percentage <1% can improve the mechanical properties of cement-based mixtures [35]. Brown seaweed that is incorporated into concrete improves the mechanical properties of the mixture for an optimal percentage of 8% [36]. To achieve a low carbon impact material, seaweed is incorporated into loamy soil and clay mixtures [1,16]. The addition of seaweed in the form of 1 cm fibers to a loamy soil mixture improves the mechanical, physical, and hydric properties. The mechanical properties will be improved by the favorable effect of seaweed fibers, which increase the strength of the soil against crack growth [37]. The brown seaweed powder of *Laminaria digitata* incorporated in clay-based soil mixtures improves the mechanical, hydric, and thermal properties [38]. In this study, the stabilizing effect of the seaweed *Laminaria digitata* is attributed to the ionic bridging between the high sodium

alginate in the seaweed (29.3%) and the divalent calcium cations from the soil, as reported by [38].

In this study, we propose to better understand the following:

- The effect of two different fibers on the variability of earth-based materials' mechanical and hygrothermal characteristics (compressive strength, vapor permeability, thermal conductivity, specific heat capacity, and desorption isotherms). For this purpose, we studied flax and algae fibers.
- The influence of the algae fiber content on the different cob properties.
- The effect of algae incorporation in cob on buildings energy needs.

To do so, samples were produced with three different seaweed rates (1, 2.5, and 4% by weight) and with 2.5% flax straw for the traditional cob. These ratios enabled us to maintain the usual proportions of the cob construction, i.e., the fiber content of a fresh cob between 20 and 30 kg/m$^3$ [17]. The mechanical, thermal, and hydric tests were carried out on cylinders of Ø110 × 220 mm for the compressive strength test, and on prismatic panels of 200 × 200 × 40 mm$^3$ for the thermal tests (thermal conductivity and specific heat capacity); on cubic samples measuring 30 × 30 × 30 mm$^3$ for the sorption/desorption test, and Ø110 × 220 × 30 mm$^3$ for the water vapor permeability test.

Earthen construction has been a prevalent method of building for centuries, using locally sourced materials to create durable and sustainable structures. However, with the increasing concern for environmental sustainability, there is a growing interest in exploring alternative building techniques that can reduce the carbon footprint and promote ecological balance. Algae have recently emerged as a promising solution in the construction industry due to their unique ability to sequester carbon. Incorporating algae in earthen construction is a relatively new approach that has gained attention for its potential to enhance the hygrothermal and mechanical performance of buildings while reducing their impact on the environment. Additionally, algae can absorb heat, thus reducing the amount of energy needed to cool a building during hot weather. In this article, we examine the effect of incorporating algae in earthen construction and present the results of two simulations, one with and one without algae using Pleiades tools.

## 2. Materials and Methods

### 2.1. Materials and Samples Preparation

#### 2.1.1. Algae Fibers

The type of stranded algae that was examined in this study was *Sargassum muticum*, which is an invasive brown algae that harms ecology, fishing, and leisure activities [39]. It has also been linked to changes in biodiversity because it competes with native algae for nutrients, light, and space.

Wild *Sargassum muticum* macroalgae were collected from the Normandy coast (Ouistreham) during low tide. The sargassum fibers were dipped in the CaCO$_3$ solution (pH 11) for 6 h to eliminate chloride. The fibers are washed with tap water, dried in oven 80 °C for 30 min, and then air-dried for 1 week.

Before being cut, raw *Sargassum muticum* algae were dried in an oven at 40 °C for at least 24 h (Figure 1). Seaweed fibers have a length of 70 ± 10 mm. Fresh algae have a water content of about 64 wt% and 10.9 wt% after drying. The *Sargassum muticum* became green. This phenomenon is attributed to a greater oxidation of algal pigments with temperature [40]. The density of the *Sargassum muticum* after drying was 1362 kg/m$^3$.

Using a nitrogen-to-protein factor, the average protein content of *Sargassum muticum* was found to be 9.9 ± 1.4%. The Kennedy and Bradshaw technique [27] was used to determine the amount of *Sargassum muticum* alginate. The basic idea is to precipitate alginate using polyhexamethylbiguanidium chloride. Alginate constitutes for 12.9 ± 0.9% of the dry weight of the biomass in the *Sargassum muticum*. ICP-OES (ICP equipped with optical emission spectroscopy) was utilized to measure Ca, Na, K, and Mg. Perkin Elmer Optima DV4300 was used to determine the metal content of *Sargassum muticum*. In this

order, the four most abundant metal cations in *Sargassum muticum* are: Ca, K, Na, Mg, Fe, Al, Sr, As, Ba, and Zn follow, with traces of Cu, Cr, Ni, Pb, and Mo (Table 1).

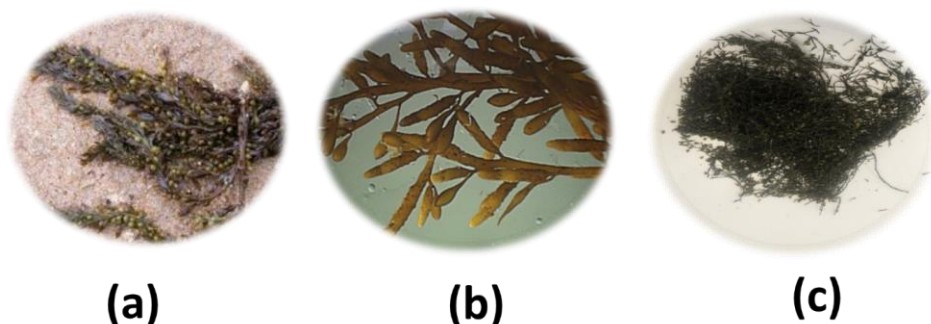

**Figure 1.** Processing of *Sargassum muticum*. (**a**) Collection; (**b**) washing; and (**c**) drying with color change.

**Table 1.** Chemical composition of *Sargassum muticum* algae.

| Element | Al | As | Ba | Ca | Cu | Cr | Fe | K | I | Mg | Mo | Na | Ni | Pb | Sr | Zn |
|---|---|---|---|---|---|---|---|---|---|---|---|---|---|---|---|---|
| mg/kg | 413.7 | 53 | 23.1 | 56936 | 2.3 | 0.4 | 811 | 30056 | 5670 | 8763 | 0.5 | 11813 | 0.4 | 0.2 | 332 | 7 |

### 2.1.2. Flax Straw Fibers

As a traditional cob, we used flax straw fibers as reinforcement materials. These fibers present various advantages because of their wide availability and low cost. Therefore, their utilization might lead to a reduced environment impact as compared to the polymeric fibers due to their renewability, biodegradability, and neutrality of $CO_2$ emissions. The flax straw utilized in this research was provided by local farmers (Laulne, Normandy). The density of the flax straw was 1408 kg/m$^3$. Flaw straws have a water content of about 4 wt%. The biochemical composition of flax straw shows the presence of cellulose (52.3%), hemicellulose (26.4%), lignin (14.7%), ash (5.1%), and extractives (1.5%). The chemical composition of flax straw is presented in Table 2.

**Table 2.** Chemical composition of flax straw.

| Element | Al | As | Ca | Cu | Fe | K | Si | Mg | P | Mn |
|---|---|---|---|---|---|---|---|---|---|---|
| mg/kg | 243.1 | 0.6 | 1720 | 1.8 | 254 | 10468 | 35347 | 619 | 689 | 35 |

The specific gravity, the tensile strength, and the Young's modulus of each fiber are presented in Table 3.

**Table 3.** Physical properties of fibers used in this study.

| Fiber | Specific Gravity | Tensile Strength (MPa) | Young's Modulus (GPa) |
|---|---|---|---|
| Flax straw fiber | 1.4 | 23.9 | 57 |
| *Sargassum muticum* fiber | 1.3 | 21 | 25 |

For the structural cob wall, we used a mixture of silty soil and sand. The soils were collected in Lieusaint carry (Sociétés des Sablières du Cotentin, SABCO Normandy). The classification of soils is performed using traditional geotechnical classification and according to the applicable standards. The particle size distribution particle was performed differently for fine particles (laser particle sizing with LS 13 320 from Beckman Coulter)

and coarse particles (>80 µm) according to XP P94-041 [41]. In addition, the soil plasticity is determined by the Atterberg limits according to the standard NF P94-051 [42]. The clay is determined by the value of methylene blue (MBV) according to the NF P94-068 standard [43]. Atterberg limits are used to determine the liquid limit (LL), plastic limit (PL), and plasticity index (PI). The value of methylene blue and the particle size classify the soil as sandy, silty, or clayey [44].

Results of the characterization tests are indicated in Figure 2 (Atterberg limit and MBV) and shown in the Table 4.

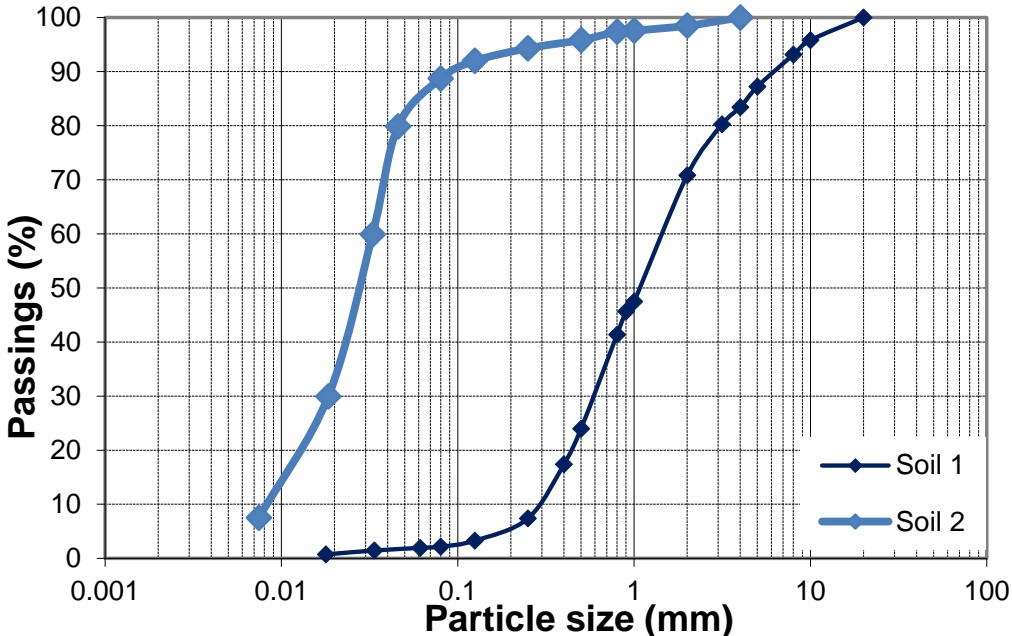

**Figure 2.** Soil particle size distribution after crushing.

**Table 4.** Atterberg limits and methylene blue values of the crushed soil.

| Parameter | LL (%) | PL (%) | PI (%) | MBV (g/100 g) | USCS (5) |
|---|---|---|---|---|---|
| Soil 1 | 28.2 | 23.1 | 3.7 | 1.42 | Low plasticity silt |
| Soil 2 | 22.9 | 20.8 | 2.4 | 0.79 | Silty sand with gravel |
| Soil 3 | 24.5 | 28.1 | 4.2 | 5.34 | Silty clay |

LL: liquid limit; PL: plastic limit; Pi: plastic index; and MBV: methylene blue value.

Soil, 1 which was utilized for the structural cob, was procured in Lieusaint Carry (Normandy). Silicon, aluminum, iron, and sodium constitute the major cations in the structural earth that were employed in this investigation (Table 5). The mineralogical analysis shows that soil 1 is composed of the following main major phases: quartz $SiO_2$ (54.8%), muscovite $KAl_2(AlSi_3O_{10})(OH, F)_2$ (26.2%), montmorillonite $(Na, Ca)_{0,3}(Al,Mg)_2Si_4O_{10}(OH)_2 \cdot nH_2O$ (6.9%), and albite $NaAlSi_3O_8$ (4.2%), with minor occurrences of Kaolinite, illite, goethite, huntite, and rutile [45]. With a rate of 6.9%, montmorillonite is the only expandable species discovered in our soil. The proportion of muscovite, albite, kaolinite, and illite in our soil influences its shrinkage characteristics. These crystals have little water between their layers because of their limited interfoliar area [46]. As a result, when immersed in water, they exhibit little intercrystalline swelling [47]. When dried, these four species shrink significantly less than smectite clays such as montmorillonite [48]. Quartz was the first weather-resistant mineral to form. Other minerals found in trace amounts include feldspars, mica, limonite, and iron oxy-hydroxides [49]. Finally, silty soil is typical of the soil employed for structural

cob. It is primarily made up of unaltered minerals such as quartz and silicates grains (smectites, feldspars, micas, and serpentines). The density of soil 1 is 2630 kg/m$^3$.

**Table 5.** Composition of soil 1 in weight obtained by EDS measures.

| Element | Wt.% |
|---|---|
| $SiO_2$ | 65.89 |
| $Al_2O_3$ | 14.23 |
| $Fe_2O_3$ | 6.65 |
| MgO | 4.08 |
| $K_2O$ | 2.17 |
| $TiO_2$ | 2.08 |
| CaO | 1.27 |
| $Na_2O$ | 1.11 |
| MnO | 0.16 |
| $P_2O_5$ | 0.14 |
| $SO_3$ | <0.1 |
| Loss on ignition | 2.26 |

Soil 2, which was used in the structural cob, is a natural French sand. It is composed of natural quartz (99% of $SiO_2$). The density of soil 2 is 2606 kg/m$^3$.

### 2.1.3. Water

The local tap water was used to produce all cob specimens. The chemical composition of the water used is listed in Table 6. The pH value of water is 7.69.

**Table 6.** Chemical composition of tap water used in this study.

| Element | Concentration in mg/L |
|---|---|
| Ca | 88.7 |
| Mg | 38.4 |
| Na | 42.9 |
| Cl | 52.0 |
| K | 9.1 |
| P | 2.6 |
| F | 0.3 |
| Zn | 0.2 |
| Fe | <0.1 |
| Cu | <0.1 |
| Mn | <0.1 |
| $SO_4^-$ | 8.4 |
| $NO_3^-$ | 5.2 |

### 2.1.4. Mix Design

The soil used in our study is composed of 2/3 soil and 1/3 sand. Different mixes were prepared. The flax and the seaweed fibers were cut to the length $70 \pm 10$ mm. These fibers were randomly inserted and blended until a homogeneous composite was obtained.

We prepared 3 mixtures of cob specimens with different percentage of seaweed fibers (1, 2.5, and 4%) and compared with the standard cob (2.5% of flax straw).

The composition of the mixture is presented in Table 7. After blending, the cob samples were stuffed into the cylindrical molds (diameter Ø110 mm × height 220 mm) for mechanical testing (according to EN 1015–11 standard) and into the Prismatic molds (20 × 20 × 4 cm$^3$) for thermal testing (Figure 3).

**Table 7.** Composition of the mixture.

| Formulations | Silty Soil (kg) | Sand (kg) | Seaweed Fiber (kg) | Flax Fiber (kg) | Water (kg) |
|---|---|---|---|---|---|
| R0 | 21.84 | 10.92 | - | 0.82 | 6.71 |
| R1 | 21.84 | 10.92 | 0.33 | - | 6.62 |
| R2.5 | 21.84 | 10.92 | 0.82 | - | 6.71 |
| R4 | 21.84 | 10.92 | 1.33 | - | 6.81 |

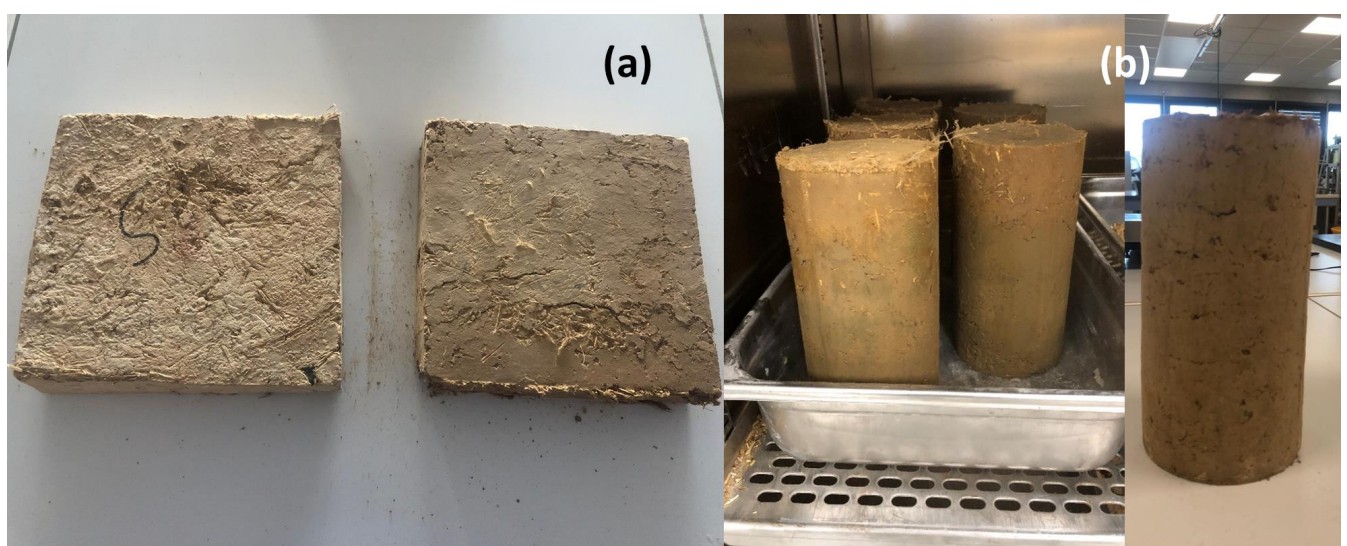

**Figure 3.** Structural cob samples used for (**a**) thermal tests and (**b**) mechanical tests.

The soil and seaweed fibers or flax straws are first blended dry by hand. The necessary amount of water is added and blended with the soil and seaweed mixture until the mixture is liquid. The fibers are then added gradually to the PROVITEQ mixer for 120 s. The mixture is packed by hand and stocked at regular temperature ($20 \pm 1\ °C$) for 24 h. After 24 h, the mixture was then manually compressed in oiled molds. The appropriate weights of all mixtures were determined in order to obtain the same density. For each specimen, we needed approx. 1800 kg/m$^3$ of dry density.

The mixtures are placed at room temperature and are covered with a plastic bag for a duration of 24 h and then we put the mixtures in cylindrical and prismatic molds. Then, we put them in a climatic chamber at a temperature of $20 \pm 1\ °C$ for 48 h and then we put them in the oven at a temperature of $40\ °C$. The drying process takes approximately 21 days. For each selected composition, four replicates were prepared. After the drying period, the samples were placed in a climate chamber under ambient conditions ($20 \pm 1\ °C$ and 50% relative humidity). It is important to note that no cracking was observed, and the shrinkage was still very low (<1%) for all cob samples after 28 days. Another important point to note is that no mold was detected in our samples after 6 months.

The characterizations of the specimens began once the equilibrium was achieved, i.e., when the variation in mass between two daily measurements was less than 0.1%. In general, this equilibrium was obtained after 48 h. It is important to note that no cracks were noticed, and that the shrinkage remained very small (less than 1%) for all specimens after 28 days.

The measured specific gravity of the mixture was between 1800 and 1900 kg/m$^3$. For comparison, we have prepared the standard cob insulation material corresponding to the one studied in the CobBauge project [6].

*2.2. Experiments*

2.2.1. Physical and Chemical Properties

Energy-dispersive X-ray spectroscopy (EDS) was used to assess the chemical composition of the raw components. With the use of a SUPRA™ 55 (SAPPHIRE; Carl Zeiss, Jena, Germany).

Apparent porosity is one of the main physical parameters in the characterization of materials. This parameter is determined by immerging the small specimens in a non-wetting oil; in this work, we used dearomatized oil. The analyses are realized in accordance with the NF ISO 5017 standard [50].

The samples are saturated by oil under vacuum in a desiccator for 24 h, which allows the liquid to displace the air in the pores without interacting with the sample.

Then, the samples were submerged under oil and then measured in air. The specimens were dried in the oven at $105 \pm 5$ °C until a constant mass was obtained, i.e., until two successive measurements, before and after a 24 h stay in the oven, did not differ by more than 0.01%, which was then considered the dry mass.

This makes it possible to obtain the mass that was initially filled with oil, by means of a difference in mass between saturated and dry state. Thus, the accessible porosity $P_0$ is given by the following:

$$P_0 = \frac{M_{air} - M_{dry}}{M_{air} - M_{oil}} \tag{1}$$

where:

- $M_{dry}$ is the mass of dry specimen.
- $M_{air}$ is the mass of saturated specimen in air.
- $M_{oil}$ is the mass of saturated specimen in oil.

2.2.2. Mechanical Properties

The mechanical strength of cob should be adequate to construct multi-story buildings. The main mechanical parameter of cob is its compressive strength. To examine how the incorporation of *Sargassum muticum* affects the mechanical performance of cob, unconfined compressive strength (UCS) tests have been applied. These tests are performed in accordance with EN 1015-11 [51]. The samples have a diameter of 11 cm and a height of 22 cm. The compressive strength at 1% shrinkage is presented in this study because this value is more representative of the building's behavior [52].

The tests were realized under a regulated force. The stress/strain curve is obtained from the displacement and force measurements made by the sensors incorporated in the press. Three samples of each formulation were prepared and tested. The UCS was determined with an INSTRON SCHENCK press with a capacity of 100 kN.

2.2.3. Thermal Properties

Thermal conductivity was investigated using a NETZSCH thermal flow meter (model HFM 446 Lambda). The sample, with parallelepiped dimensions of $20 \times 20 \times 4$ cm$^3$, is set between the cooling and heating plate; heat flows from the heating plate through the sample to the cooling plates from which it is extracted. The cooling and temperature of the hot plate are adjusted by a Peltier cryostat to create a temperature gradient from the hot plate to the sample on the cooling plate from $-10$ to 40 °C.

The specific heat capacity (Cp) is a major parameter indicating the ability of a material to store heat. The test consists of scanning the temperature and tracking the heat flow into or out of the investigated material relative to a reference. In this study, the specific heat capacity of the specimens is tested between 0 °C and 40 °C. The heating rate is set at 1 °C/min.

2.2.4. Hygroscopic Properties

In this study, the sorption isotherms were measured according to the discontinuous method: The moisture content was determined at successive stages of increasing relative humidity (RH). The specimens were dried in an oven at 60 °C until the mass change was less than 0.1% between three consecutive weightings with 24 h time-step. Then, the specimens were stabilized in a climate chamber (Memmert HPP260), which regulates temperature and relative humidity. The specimens were weighted two to three times a week. The sorption isotherm was measured at $23 \pm 0.1$ °C. The technique of dynamic vapor sorption (DVS) was used according to ISO 12571 standard. Relative humidities considered in this study were 0, 35, 50, 65, 80, and 90% [53]. The water content W was calculated from the mass of the specimen with equation.

$$W = (m - m_0)/m_0 \tag{2}$$

where m is the mass of the specimen in steady state conditions (kg) and $m_0$ is the mass of the specimen in the initial dry state.

In this study, the selected model was the GAB one. This model relates the water content to the specific surface area of the material for multilayer sorption.

The GAB model covers a wide range of RH (from 5 to 80–90% RH) and is convenient to fit experimental adsorption data all over the RH range:

$$W/Wm = \frac{Cg \cdot k \cdot RH}{(1 - k \cdot RH)(1 - k \cdot RH + Cg \cdot k \cdot RH)} \tag{3}$$

where Wm is the monomolecular water content (kg/kg), Cg and k are the fitting parameters of the GAB model, and RH is the relative humidity.

The fitting of the model to the experimental values was performed using a least-squares minimization procedure.

In order to study the capacity of a material to let steam through, it is important to know the water vapor permeability. This parameter gives the ratio of the quantity of water vapor passing through a material per unit of the time, thickness, and difference in vapor pressure existing on each side of the material.

In the present study, the water vapor permeability of the different specimens was measured according to the ISO 12572 standard [54]. This measurement is taken by tracking the mass of a specimen subjected to a moisture gradient. The dry cup technique requires the creation of a moisture gradient from inside the cup ($\approx$0% RH) to outside ($\approx$50% RH) through the sample.

The measurements of water vapor permeability by the dry cup method provide information on the behavior of the material when moisture transfer is dominated by vapor diffusion. The vapor flux through the material is obtained by weighting the sample cup assembly. In steady state, the water vapor flux (G) through the sample is given by the slope of the regression line of the sample cup assembly mass versus time. This result is obtained after removing the initial non-linear phase of the test. Thus, the vapor surface flux density (g) (or transmission rate) is determined using the area A of the exposed sample.

$$g = \frac{G}{A} \tag{4}$$

The water vapor resistance (Z) can be determined from the water vapor permeance (W). The latter is dependent to the partial pressure of water vapor $\Delta Pv$ between the sides of the specimen.

$$Z = \frac{1}{W} = \frac{A \times \Delta Pv}{G} \tag{5}$$

Then, the water vapor resistance of the air layer in the measurement cup (Za) is obtained by finding the thickness of the air layer (da) and the water vapor permeability of the air at atmospheric pressure (δa), which is equal to $2 \times 10^{-10}$ kg.m$^{-1}$.s$^{-1}$.Pa$^{-1}$.

$$Za = \frac{da}{\delta a} \tag{6}$$

The corrected water vapor permeance Wc is obtained using this equation:

$$Wc = \frac{1}{Z - Za} \tag{7}$$

Second, the water vapor permeability of the specimen (δ) can be determined using the following Equation (8). e is the thickness of the specimen.

$$\delta = Wc \times e \tag{8}$$

Finally, the water vapor resistance factor of the specimen (μ) is found by the following equation:

$$\mu = \frac{\delta a}{\delta} \tag{9}$$

The moisture levels required for testing were generated by silica gel (≈0% RH) and magnesium nitrate (≈50% RH). All specimens were preconditioned at 23 °C and 50% relative humidity.

Figure 4 shows the experimental tests conducted in this study.

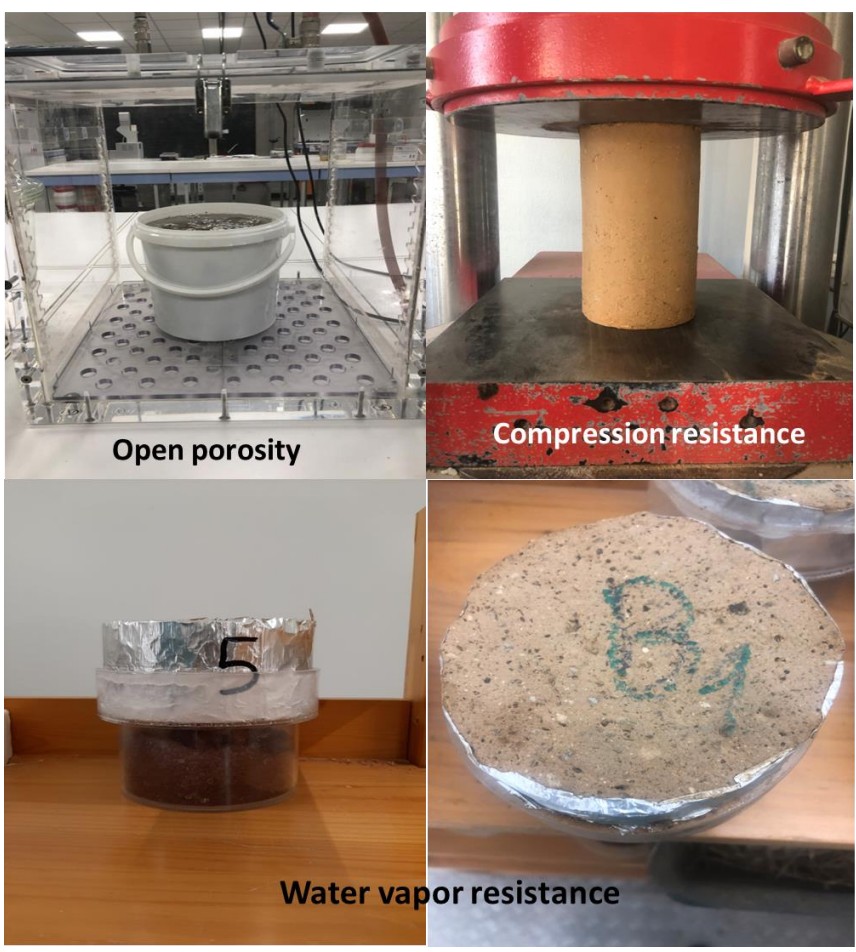

**Figure 4.** Porosity, mechanical, and water vapor resistances tests conducted in this study.

## 3. Results and Discussions

### 3.1. Physical and Mechanical Properties

The density is almost equal in all the mixtures with different seaweed contents and is close to that of the standard cob (Figure 5). The open porosity increases with the rate of seaweed incorporated. An increase of 20% between the mixture with 1% seaweed and the mixture with 4% is observed., this increase can be explained by the fact of absorption due to the loss of water, thus reducing the volume of the seaweed fibers during the drying process compared to that of the flax straw. Another explanation for this increase is that flax straw bends and clumps in the mixture, whereas seaweed fibers are thin and break in the mixture, which explains the increase in pores when the fibers are added at a long distance. The high presence of seaweed increases the porosity. This high porosity causes the drop in mechanical strength.

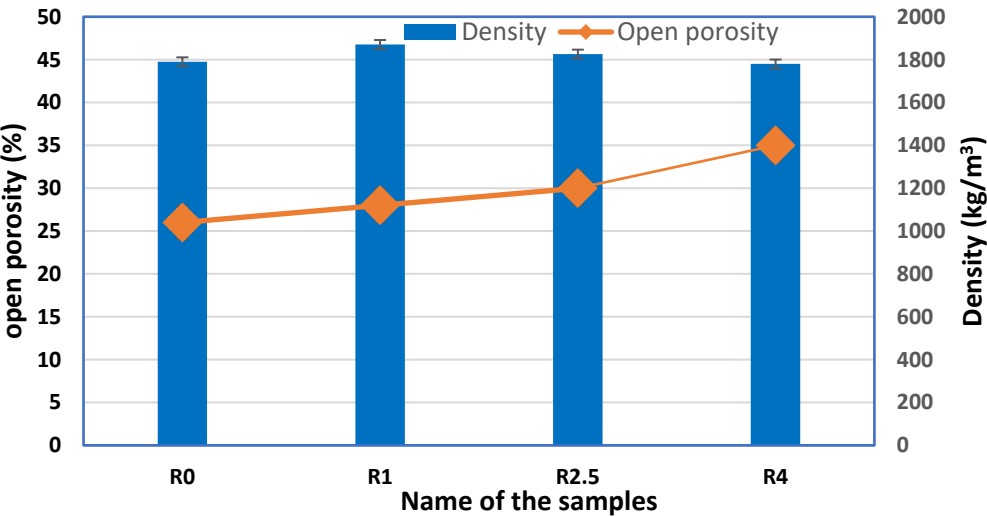

**Figure 5.** Cob's density and open porosity variation versus *Sargassum muticum* content.

Figure 6 shows the optical microscope images of cob specimens and thus the difference in porosity for the four formulations (R0–R1–R2.5–R4). The results show the increase in the porosity with the presence of seaweed and with the increase in the amount of fiber used. Natural fibers are known to have varying porosity because they shrink and swell when they are filled into molds [55]. Additionally, natural fibers may shrink as a result of water saturation, which increases porosity and reduces water flow resistance in saturated flows.

In the compression testing, no grain breakage was seen in addition to the lack of adhesion between the mix's components. Therefore, it is impossible to determine the compressive strength of these ductile materials using the highest stress ever recorded [56]. The maximum compressive strength over a range of longitudinal deformation percentages ranging from 1 to 7.5% has been determined in recent investigations [56]. We selected the appropriate compressive strength at a 1% strain for this study. This difference between the seaweed samples and the standard cob may be due to the ductility of the flax, which is greater than that of the seaweed and porosity.

The incorporation of seaweed results in a decrease in the mechanical strength of the cob (Table 8). This could be due to the addition of seaweed, which increases the pore's quantity in the mixtures and subsequently, the percentage of the open porosity. Nonetheless, in this work, the weakest measuring compressive strength (R4) is equal to 0.81 MPa. This value is regarded as being adequate to construct a two- or even three-storey dwelling [31,32].

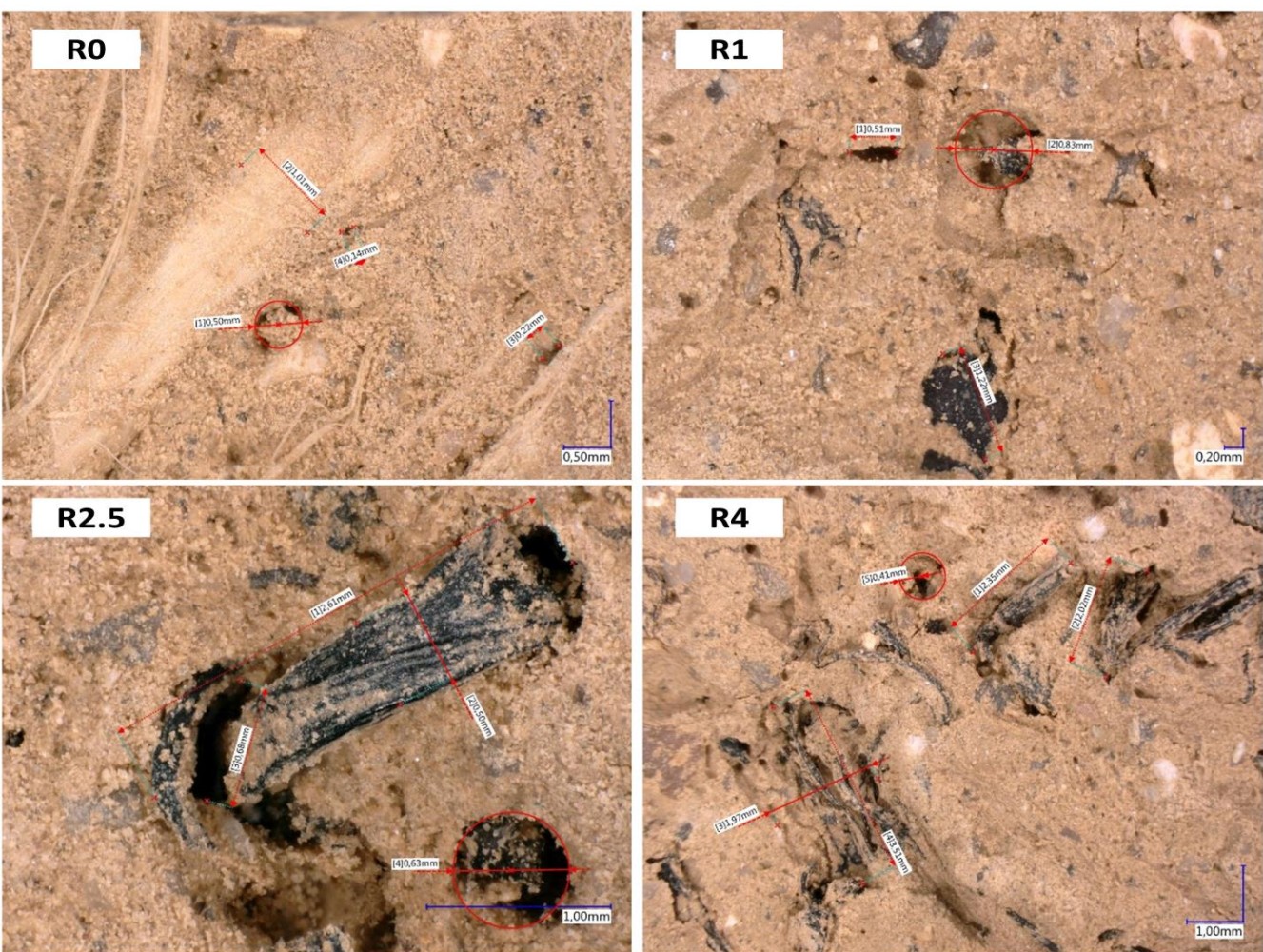

**Figure 6.** Microscopic images of the four formulations.

**Table 8.** Maximum values of cob's compressive strength versus *Sargassum muticum* seaweed content.

| Samples | R0 | R1 | R2.5 | R4 |
|---|---|---|---|---|
| Compressive strength (MPa) | 1.66 | 1.33 | 1.25 | 0.81 |

### 3.2. Hygroscopic Properties

The value of the water vapor resistance factor ($\mu$) of the R0 sample (Figure 7) is fairly similar to that mentioned by Phung ($7 < \mu < 10$) [57].

Additionally, it can be observed that the water vapor permeability increases proportionally with the seaweed content. The drop in the water vapor permeability, equal to 7%, 15%, and 20% for R1, R2.5, and R4, correspond to a cob gain ability to allow moisture to pass through thanks to enhanced porosity. This increase in permeability may be correlated with the increase in porosity. Furthermore, since *Sargassum muticum* exhibits hydrophilic traits, this characteristic could also affect the hygroscopic behavior of the cob, including its water vapor permeability.

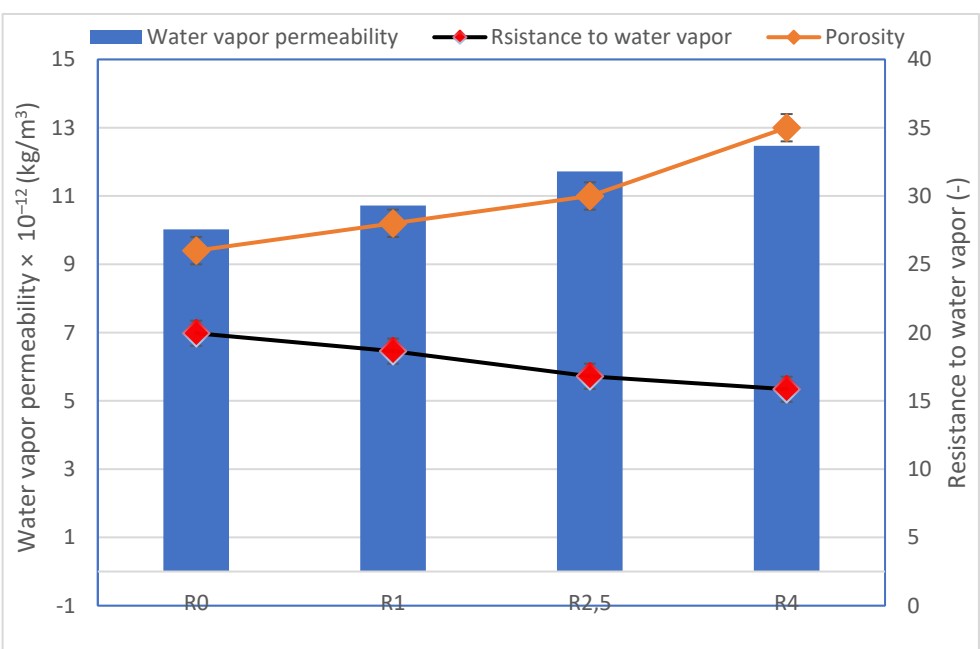

**Figure 7.** Water vapor permeability and resistance factors measured for all the formulation mixtures compared to cement mortar [58].

The moisture sorption isotherms at increasing RHs in ambient air at a temperature of 23 °C (Figure 8) show that adding seaweed to the cob enhances the quantity of the absorbed moisture.

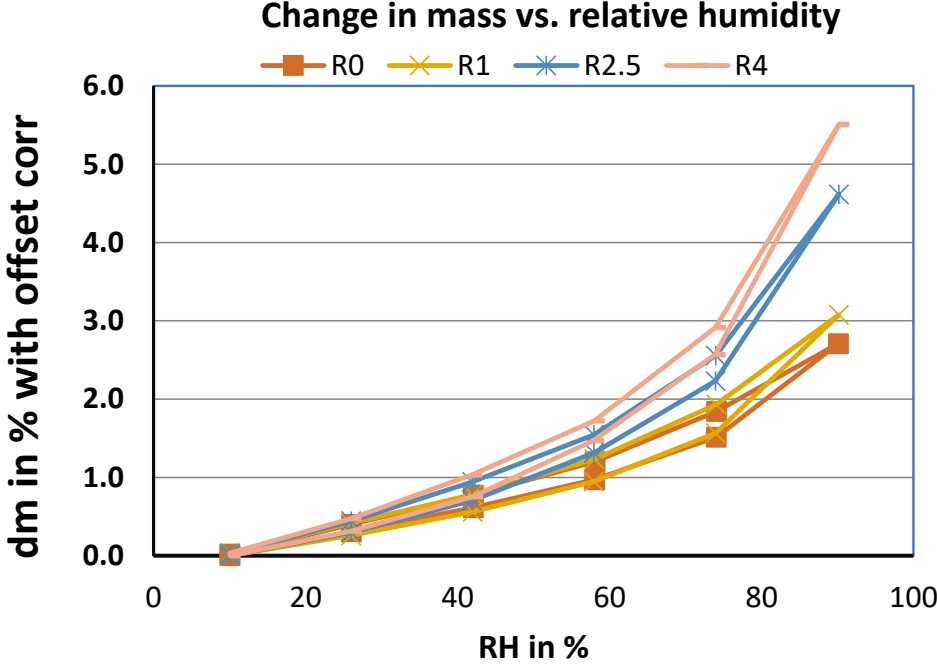

**Figure 8.** Moisture sorption isotherm of cob with different seaweed contents.

This behavior comes from the hydrophilic nature of seaweed. For instance, at a RH value of 90%, the absorbed moisture in the cob reaches 2.5% of the sample mass. The water vapor permeability at 90% RH is equal to 17, 44, and 55% for R1, R2.5, and R4, respectively. Furthermore, by looking at the sorption and desorption curves, a hysteresis can be noticed. According to the IUPAC (International Union of Pure and Applied Chemistry), such

hysteretic loops curves are classified as Type II-isotherms. The Type II-isotherm is observed in porous materials such as the soil–vegetable fiber mixes investigated in this research. The results show an increase in the mass differential with the incorporation of seaweed. This phenomenon is caused by a number of reasons, including the increased porosity and the high absorbance of algae.

The high absorbed moisture of *Sargassum muticum* is attributed to the presence of organic components such as alginate. Alginate is a polysaccharide hydrocolloid that serves as an important hydrophilic element due to its ability to absorb and retain a significant amount of water within its structure [59,60].

This means that cob and seaweed mixtures can release all the absorbed water vapor and return to their original state. Finally, the addition of seaweed has largely increased the moisture content of the cob, but these characteristics are still comparable to those of other earth-based materials, both in terms of the moisture sorption and moisture vapor permeability. However, when added to the soil–water mixture, *Sargassum muticum* alga will prevent surface condensation (mildew) due to its high sorption capacity.

As shown in Figure 9, the thermal conductivity (λ) behavior of the cob–seaweed mixtures is measured at four different temperatures (0 °C, 10 °C, 20 °C, and 30 °C).

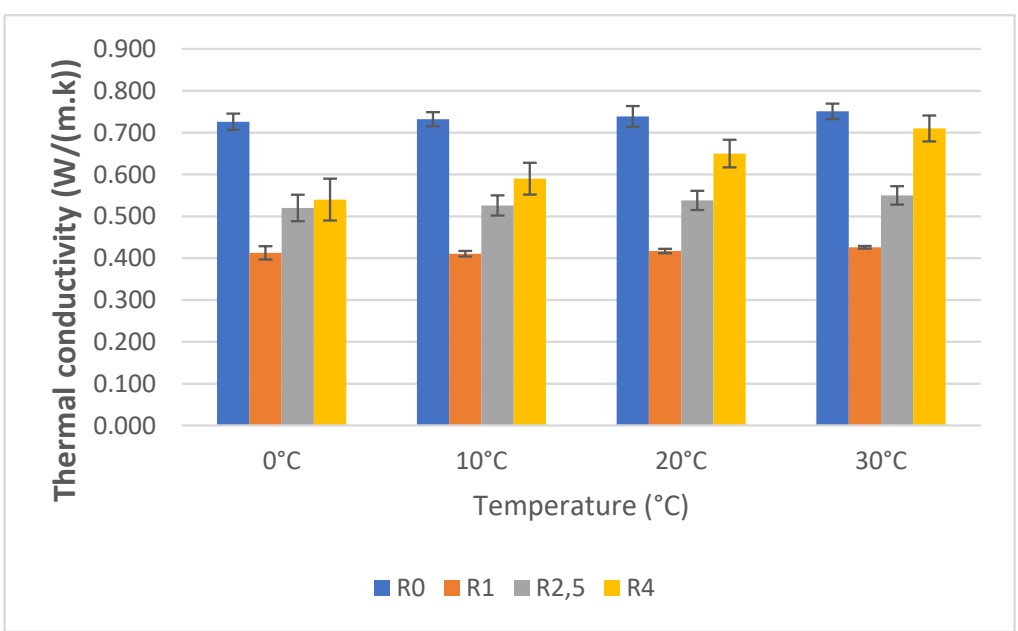

**Figure 9.** Thermal conductivity of cob compositions as a function of the temperature and the *Sargassum muticum* content.

The results show a decrease in the value of thermal conductivity from 0.7 W/(m.K) for the standard cob to reaches 0.413 W/(m.K) for the formulation with 4% of seaweed fibers. These values approach those reported by Collet-Foucault et al. and Hamdaoui et al. [61,62], i.e., between 0.4 and 0.7 W/(m.K).

This decrease upon seaweed incorporation is assumed to be due, on one hand, to the low thermal conductivity of the *Sargassum muticum*.

On the other hand, the thermal conductivity of dry earth depends primarily on its density and porosity. Furthermore, this reduction in the thermal conductivity is related to the decrease in density. Some previous studies [59] have revealed the relationship between thermal conductivity and the density of a material. Since a porosity increase is observed upon algae content, we contribute the major λ-decrease to this porosity enhancement.

The thermal conductivity stabilized with 4% seaweed decreases by 21% when compared to the mixtures stabilized with 1% seaweed.

These results provide indicators of the optimal amount of *Sargassum muticum* to be incorporated. The optimal content of *Sargassum muticum* must be determined according to the different properties and nature of the materials. This rate is defined by the thermal properties (thermal conductivity and specific heat capacity) and hydric properties (water vapor permeability and sorption/desorption behaviour).

The thermal and hydric results show that the addition of 4% *Sargassum muticum* has the lowest thermal conductivity (0.413 W/(m.K)) and largest water vapor permeability ($12.72 \times 10^{-12}$ kg/m$^3$). However, the addition of 1% of *Sargassum muticum* has the best results in terms of the compressive strength (1.33 MPa) compared to 2.5 and 4% of seaweed addition.

Otherwise, it is found that the temperature does not noticeably affect the thermal conductivity of a cob incorporating seaweed fibers.

The evolution of the specific heat capacity of the mixtures as a function of temperature is shown in Figure 10. As expected for such materials, the specific heat capacity is constant for the seaweed content, ranging between 1% and 4%. Even though the seaweeds present a significant Cp value [38], their low addition into the soils could not influence the Cp value of the formulations.

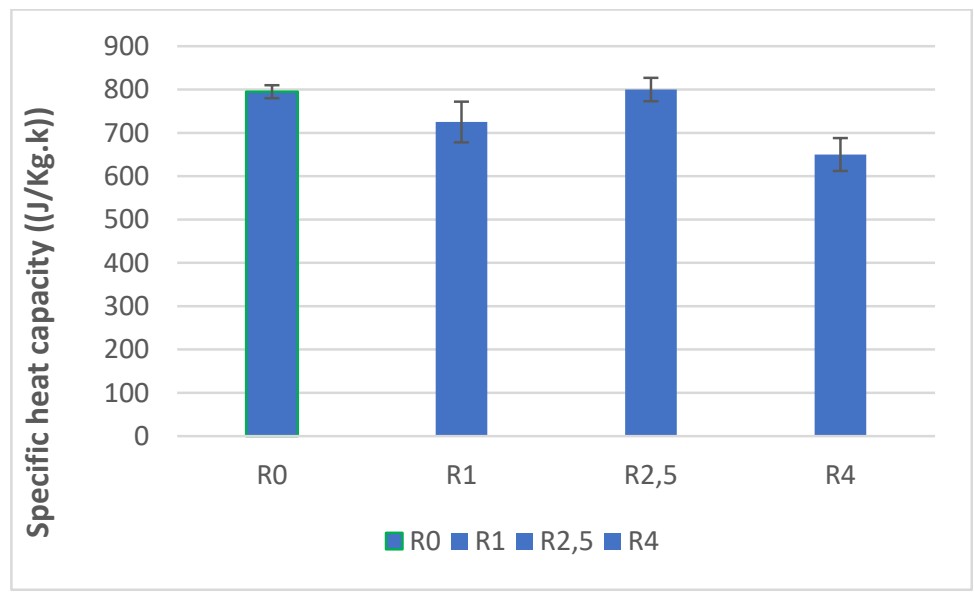

**Figure 10.** The variation of the specific heat capacity of cob as a function of *Sargassum muticum* content.

With the increasing body soil matrix porosity, the cob's insulating capacity increases. The pore formation in the soil body was enhanced by the organic additions, such as algae, and the cob showed a high insulating performance with acceptable mechanical qualities. By limiting the heat transfer and enhancing the thermal comfort, the use of algae in cob building improves its thermal efficiency. It makes sense to link the environmental and energy considerations to the effectiveness of the cob thermal analysis. The energy needed to heat or cool an area is much higher in buildings with less insulation. Consequently, a structure composed of cobs coupled with algae will require less fuel and energy to maintain the occupant's thermal comfort.

### 3.3. Comparison with Literature

The results of this study are compared with earlier studies on the hydrothermal performance of materials made of earth–fiber mixtures. The reported thermal conductivities, densities, specific heat capacities, and maximal isotherm sorption are shown in Table 9.

**Table 9.** Hygrothermal properties of different soil–fiber mixtures.

| Composition | Density (kg.m$^{-3}$) | Conductivity (W.m$^{-1}$.K$^{-1}$) | $C_p$ (J.kg$^{-1}$.K$^{-1}$) | Sorption Max (%) | Reference |
|---|---|---|---|---|---|
| Silty soil (0–4% algae) and 2% wheat straw | 1780–1871 | 0.4–0.7 | 650–795 at 20 °C | 2.7–5.5 | Present study |
| Silty soil (0–10% phase change materials) and 2.5% wheat straw | 1400–1500 | 0.64–0.76 | 1164–3862 at 24 °C | 2–2.5 | Alassaad et al., 2022 [52] |
| Silty soil and 2.5% hemp straw | 1107–1583 | 0.75 | 1205 at room temperature | - | Zeghari et al., 2021 [63] |
| Silty soil and 2.5% wheat straw | 1107–1583 | 0.32 | 1205 at room temperature | - | Zeghari et al., 2021 [63] |
| Silty soil and 2.5% flax straw | 1107–1583 | 0.42 | 1205 at room temperature | - | Zeghari et al., 2021 [63] |
| Silty soil and 2.5% reed | 1107–1583 | 0.36 | 1205 at room temperature | - | Zeghari et al., 2021 [63] |
| Silty soil and 0–3% wheat–flax straw | 1462–2011 | 0.62–1.93 | - | 2.8–4.1 | Phung 2018 [64] |

The hygroscopic values show a wide range of fluctuation as well. The thermal conductivity values range from 0.32–0.33 W.m$^{-1}$.K$^{-1}$ [63] to 0.76 W.m$^{-1}$.K$^{-1}$ for a dry density of 1107 to 1583 kg.m$^{-3}$ [52]. This demonstrates that, among the materials with a similar composition in the literature, our mixtures present thermal conductivities that are either similar or higher. When the results are contrasted with those obtained by Zeghari et al. [63], it is evident that the Cp is low. Nevertheless, the combinations of soil and algae fibers outperform the literature's values in terms of the sorption value.

As a result, the thermal conductivity and the sorption of algae-based cob specimens are comparable or better than those of a traditional cob.

*3.4. Energy Needs*

The project involves the construction of a house built with seaweed, which is a renewable and sustainable material that is gaining popularity in the construction industry. The goal of the project was to evaluate the energy gain brought by the seaweed, particularly in terms of its insulation properties. The house was located in a coastal region (Caen city in Normandy, France) where seaweed is abundant, thus providing a local and eco-friendly source of building material. The walls of the house were made from a combination of seaweed and traditional earthen materials. The characteristics of the walls were derived from experimental characterization data. The building has a floor area of 72 m$^2$ and a total height of 3 m with seven uncoated, double-glazed windows on the building's north and south facades. The examined building's design, the materials used for its slab, roof, and walls are presented in Figure 11 and Table 10, respectively. Additionally, a representative occupation scenario was defined. In this sense, it was supposed that the housing would be unoccupied between 10:00 a.m. and 06:00 p.m. for the worked days except for Wednesday, where the inoccupation was supposed to be between 10:00 a.m. and 2:00 p.m. For the weekends, the housing was supposed to be occupied all the day. Furthermore, two levels of occupation were imposed: sleeping time (0.022 occup./m$^2$) and living time (0.016 occup./m$^2$).

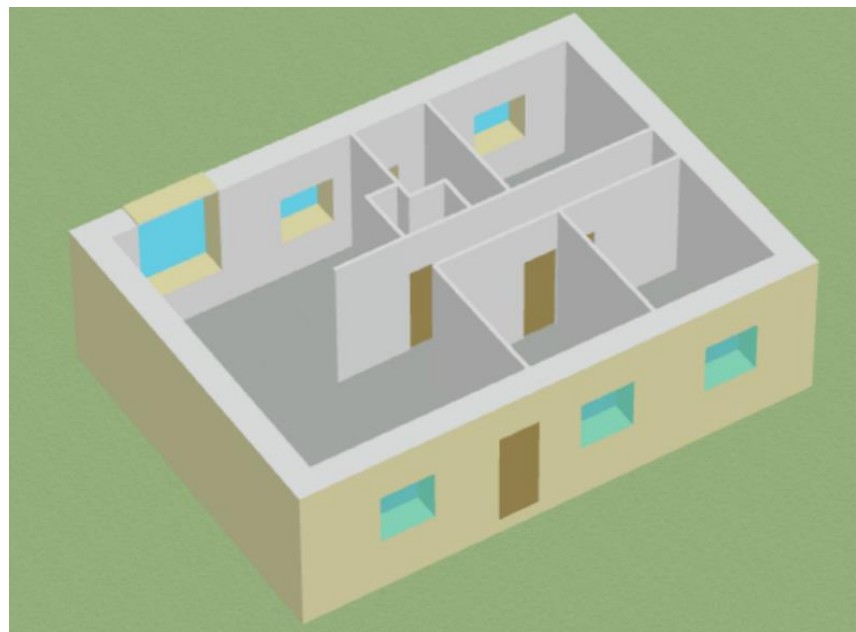

**Figure 11.** Three-dimensional plan of the studied housing.

**Table 10.** Thermal properties of the materials constituting the building envelope.

| Composition | Thickness (m) | Thermal Conductivity (W/(m.K)) | Density (kg/m$^3$) | Specific Heat Capacity (J/(kg.K)) |
|---|---|---|---|---|
| External walls | | | | |
| Flax-based cob | 0.6 | 0.739 | 1790 | 795 |
| Algae-based cob | 0.6 | 0.45 | 1826 | 797 |
| Ceiling | | | | |
| Zinc foil | 0.001 | 110.000 | 7200 | 381.6 |
| OSB | 0.012 | 0.130 | 650 | 1699.2 |
| Rock wool | 0.200 | 0.044 | 113 | 1029.6 |
| OSB | 0.012 | 0.130 | 650 | 1699.2 |
| Plaster gypsum | 0.01 | 0.400 | 1000 | 1000.8 |
| Slab | | | | |
| Tiling | 0.01 | 1.70 | 2300 | 700 |
| Mortar | 0.04 | 1.15 | 2000 | 840 |
| Concrete | 0.04 | 1.75 | 2300 | 920 |
| Hollow slab | 0.16 | 1.23 | 1300 | 648 |
| Expanded polystyrene | 0.10 | 0.039 | 25 | 1380 |

To evaluate the energy gain brought by the seaweed, we simulated energy needs to heat and cool a single-family housing built with an algae-based cob. The results are compared to those obtained in a similar house built with a flax-based cob.

The preliminary results indicate that the house built with an algae-based cob provides superior insulation, reducing the amount of energy needed to maintain a comfortable indoor temperature. The seaweed presents interesting thermal properties, regulating the temperature inside the house and reducing the need for artificial heating and cooling. The numerical results also demonstrate a 14.5% reduction in energy needs (heating and cooling), indicating that building with an algae-based cob is an effective way to improve their energy efficiency o (Figure 12).

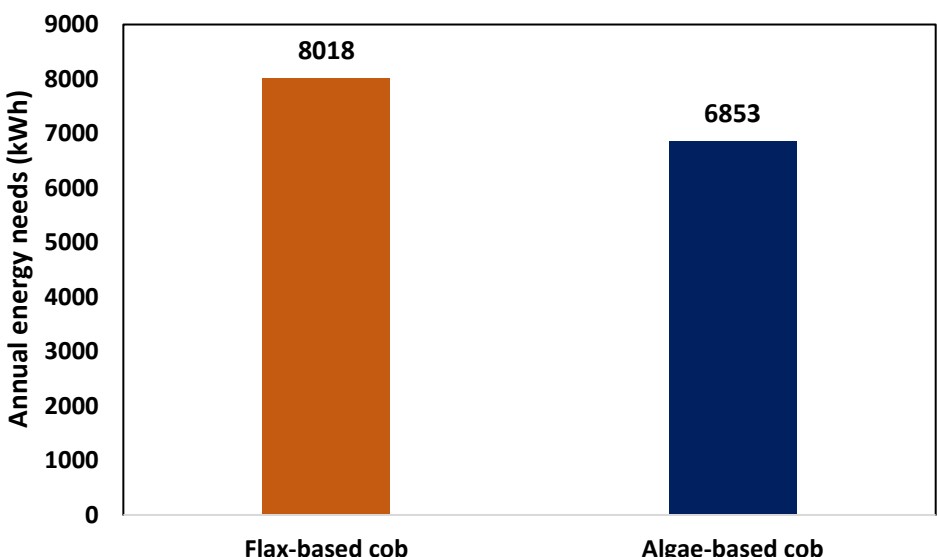

**Figure 12.** Calculation of heating and cooling needs in housing with an envelope made with algae-based cob and a flax-based cob.

## 4. Conclusions

The main objective of this research was to evaluate the mechanical and hydrothermal performances of cob walls containing seaweed fiber.

The incorporation of seaweed at different rates in soil-based mixtures according to the cob method has an influence on both the thermal and hydric comfort. Furthermore, cob incorporating 4% seaweed loses about 74% of its compressive strength when compared to cob that is incorporated with 2.5% of straw fibers. This loss is combined with ductile behavior of the cob. Additionally, the experimental studies indicated that the incorporation of *Sargassum muticum* with cob decreases its thermal conductivity by about 28%, 39%, and 44% for R1, R2.5, and R4, respectively. Therefore, the insulation ability of the mixture is increased accordingly. This decrease is due to the porosity created by the water loss by the algae during their drying process.

The numerical simulations indicated that housing with an external wall made with an algae-based cob presents 14.5% less energy requirements than equivalent housing made with a flax-based cob.

This study's findings clearly show that stranded brown algae, which are considered as waste, have the potential to enhance the hygrothermal performance of cob. Thus, the present work highlights the potential of using seaweed as a sustainable and renewable material. This may present significant benefits for buildings' energy efficiency and thermal performance. As the world continues to search for eco-friendly building materials and techniques, the use of algae in construction may provide a promising solution for a more sustainable future.

Building with an algae-based cob may present an interesting opportunity to create regional value chains that have a positive impact on the environment.

**Author Contributions:** Conceptualization, investigation, data curation, writing—original draft preparation and methodology, H.A., Y.E.M., K.T., M.-H.B. and D.C.; software, K.T. and M.-H.B.; validation, formal analysis, writing—review and editing, visualization, supervision and resources, Y.E.M., K.T. and D.C. All authors have read and agreed to the published version of the manuscript.

**Funding:** This research received no external funding.

**Informed Consent Statement:** Note applicable.

**Data Availability Statement:** The experimental and computational data presented in the present paper are available from the corresponding author upon request.

**Acknowledgments:** We thank Franck HENNEQUART (ALGAIA) for providing the *Sargassum muticum* algae.

**Conflicts of Interest:** The authors declare no conflict of interest.

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
