# Peer review of "Earth-Based Building Incorporating Sargassum muticum Seaweed: Mechanical and Hygrothermal Performances"

_buildings, doi:10.3390/buildings13040932_

Round 1

Reviewer 1 Report

-The authors spent more attention in introduction, materials and methods, whilst less attention was focused on results and discussions

-Section 3 (Results and discussions) is too weak and should be enhanced by deep analyses

-This article lacking in terms of critical discussion based on the findings obtained

-Please include more picture/figures/photos as evidences for the material and test involved

-The novelty of work is not highlighted. Why did the author choose this work instead of many other materials already in use-easily available, cost effective and eco-friendly? Is there any specific reason?

-Please explain clearly how you treated these fibers before use

-Because these fibers are organic, they can decompose and rot

**Other comments

-Lines 34-36, it is too much to cite 6 references for this sentence “It can be seen as a stand-alone natural building concept or as a collection of technologies that enhance conventional modern construction techniques with high temperature retention, durability, environmental sustainability and more methodically [3-8]”: the number of references should be decreased

-Lines 43-45, it is too much to cite 7 references for this sentence “

 As a primary material, with no additives, the earth is an excellent controller of humidity because of its capacity of absorption and desorption. It captures or release air moisture according to the ambient humidity, thus favoring a healthy indoor environment [7,12-17].”

-Line 49, please convert this reference “(Goodhew et al, 2021)”, to reference number

-Line 50, the dimensions should be in millimeter “with fiber lengths ranging from 30 to 50 cm”

-Lines 86—91, the authors should explain that the tidal zone has a deteriorate effect on building materials to inform the reader about negative effect of this zone. The authors can cite these references:

“Valorization of limestone powder as an additive for fly ash geopolymer cement under the effect of the simulated tidal zone and seawater attack”, Construction and Building Materials 369 (2023) 130616

“Effect of Tidal Zone and Seawater Attack on Alkali- Activated Blended Slag Pastes”, ACI Materials Journal, V. 119, No. 2, March 2022

“ Effect of tidal zone and seawater attack on high-volume fly ash pastes enhanced with metakaolin and quartz powder in the marine environment”, Microporous and Mesoporous Materials 324 (2021) 111261

-Line 103, please check subscript “Between 50,000 and 100,000 m3 “

-Lines 109, 114, please check “tons” OR “tonnes” in these positions and other positions

-Line 150, please explain why you used these ratios “1; 2.5 and 4%)”?

- Are these proportions by volume or by weight “1; 2.5 and 4%)”??

-Lines 152-153, all dimensions should be in millimeter “were carried out on cylinders of Ø11 x 22 cm for the compressive strength test, and on 152 prismatic panels 20 x 20 x 4 cm3 for the thermal tests“

-Line 152, why the authors used these dimensions “Ø11 x 22 cm” which did not agree with any standard

-Lines 154-155, all dimensions should be in millimeter “on cubic samples 3 x 3 x 3 cm3 for the sorption/desorption test, and Ø11 x 154 22 x 3 cm3 for the water vapor permeability test”

-Lines 154-155, why the authors used these dimensions “3 x 3 x 3 cm3 for the sorption/desorption test, and Ø11 x 154 22 x 3 cm3” which did not agree with any standard

-Lines 177-178, all dimensions should be in millimeter

-In Section 2.1, all properties of the raw materials should be included (for example, the specific gravity, tensile strength, Young’s modulus for each type of fiber should be included)

-The abbreviations included in Table 3 should be identified

-Lines 269-273, all dimensions should be in millimeter

 -In Table 6, please explain why you selected silty soil/sand ratio = 2 and did not use 2.75

-The information listed in Table 6 are vague

-In Table 6, please explain the unit for this Table. For example in R0, the silty soil is 21.84 g or kg or weight ratio or volume ratio….??/

-It is better to include Table 6 in kg/m3

-In Table 6, the ratio of fibers by weight or by volume

-Please check the title of Y-axis of Fig. 6

-Please include the error bars for Figs 3, 5, 7, 8

-Please check page 17

-Conclusions should be condensed

Author Response

We would like to thank the reviewers for their thoughtful comments and efforts towards improving our manuscript. In the following, we address comments specific to each reviewer below.

All changes in manuscript are in red color.

-The authors spent more attention in introduction, materials and methods, whilst less attention was focused on results and discussions, Section 3 (Results and discussions) is too weak and should be enhanced by deep analyses. This article lacking in terms of critical discussion based on the findings obtained

Response: This section is added to address a comparison with literature:

3.3. Comparison with literature

The results of this study are compared with earlier studies on the hydrothermal performance of materials made of earth-fiber mixtures. The reported thermal conductivities, densities, specific heat capacities, and maximal isotherm sorption are shown in Table 8.

Table 8. hygrothermal properties of different soil-fiber mixtures

Composition

Density (kg.m-3)

Conductivity (W.m-1.K-1)

Cp (J.kg-1.K-1)

Sorption max (%)

Reference

Silty soil (0-4% algae), 2% Wheat straw

1780 - 1871

0.4 – 0.7

650 – 795 at 20°C

2.7 - 5.5

Present study

Silty soil (0-10% Phase Change Materials), 2.5% wheat straw

1400 -1500

0.64 – 0.76

1164 – 3862 at 24°C

2 – 2.5

Alassaad et al., 2022 [52]

Silty soil, 2.5% hemp straw

1107 - 1583

0.75

1205 at room temperature

-

Zeghari et al., 2021 [63]

Silty soil, 2.5% wheat straw

1107 - 1583

0.32

1205 at room temperature

-

Zeghari et al., 2021 [63]

Silty soil, 2.5% Flax straw

1107 - 1583

0.42

1205 at room temperature

-

Zeghari et al., 2021 [63]

Silty soil, 2.5% Reed

1107 - 1583

0.36

1205 at room temperature

-

Zeghari et al., 2021 [63]

Silty soil, 0-3% wheat-flax straw

1462–2011

0.62–1.93

-

2.8–4.1

Phung

2018 [64]

The hygroscopic values show a wide range of fluctuation as well. The thermal conductivity values range from 0.32-0.33 W.m-1.K-1 [63] to 0.76 W.m-1.K-1 for a dry density of 1107 to 1583 kg.m-3 [52]. This demonstrates that, among materials with a similar composition in the literature, our mixtures present thermal conductivities similar or higher. When the results are contrasted with those obtained by Zeghari et al. [63], it is evident that the Cp is low. Nevertheless, combinations of soil, and algae fiber outperform literature values in terms of sorption value.

As a result, thermal conductivity and the sorption of algae-based cob specimens are comparable or better than those of traditional cob.

-Please include more picture/figures/photos as evidences for the material and test involved

Response: The Figures 3 and 4 are added as suggested by the reviewer

-The novelty of work is not highlighted. Why did the author choose this work instead of many other materials already in use-easily available, cost effective and eco-friendly? Is there any specific reason?

Response: These sentences are added in the abstract:

This study can contribute to a global environmental and economic issue, i.e. the valorization of brown algae on a large scale. Indeed, the world knows the largest sea of sargassum algae, which measures over 8850 kilometers. This huge mass of brownish algae is expanding every year, and now covers an area from Africa to the Caribbean. It weighs more than 20 million tons and extends from the Gulf of Mexico to the West coast of Africa. The findings of this study clearly show that stranded algae, which are considered as wastes, have the ability to improve the mechanical and hygrothermal performance of cob-based material.

-Please explain clearly how you treated these fibers before use, because these fibers are organic, they can decompose and rot

Response: These sentences are added in the Materials and methods section:

The sargassum fibers were dipped in the CaCO3 solution (pH 11) for 6 h to eliminate chloride. The fibers are washed with tap water, dried in oven 80°C for 30 min and then dried aired for 1 week.

It is important to note that no cracking was observed, and the shrinkage is still very low (<1%) for all cob samples after 28 days. Another important point to note is that no mold was detected in our samples after 6 months.

**Other comments

-Lines 34-36, it is too much to cite 6 references for this sentence “It can be seen as a stand-alone natural building concept or as a collection of technologies that enhance conventional modern construction techniques with high temperature retention, durability, environmental sustainability and more methodically [3-8]”: the number of references should be decreased

Response: As the reviewer suggested, the number of references is decreased

-Lines 43-45, it is too much to cite 7 references for this sentence “

 As a primary material, with no additives, the earth is an excellent controller of humidity because of its capacity of absorption and desorption. It captures or release air moisture according to the ambient humidity, thus favoring a healthy indoor environment [7,12-17].”

Response: As the reviewer suggested, the number of references is decreased

-Line 49, please convert this reference “(Goodhew et al, 2021)”, to reference number Done

-Line 50, the dimensions should be in millimeter “with fiber lengths ranging from 30 to 50 cm” Done

-Lines 86—91, the authors should explain that the tidal zone has a deteriorate effect on building materials to inform the reader about negative effect of this zone. The authors can cite these references:

“Valorization of limestone powder as an additive for fly ash geopolymer cement under the effect of the simulated tidal zone and seawater attack”, Construction and Building Materials 369 (2023) 130616

“Effect of Tidal Zone and Seawater Attack on Alkali- Activated Blended Slag Pastes”, ACI Materials Journal, V. 119, No. 2, March 2022

“ Effect of tidal zone and seawater attack on high-volume fly ash pastes enhanced with metakaolin and quartz powder in the marine environment”, Microporous and Mesoporous Materials 324 (2021) 111261

Response: As recommended by reviewer, these references are added in the manuscript

-Line 103, please check subscript “Between 50,000 and 100,000 m3 “ Done

-Lines 109, 114, please check “tons” OR “tonnes” in these positions and other positions Done

-Line 150, please explain why you used these ratios “1; 2.5 and 4%)”?

Response: The fiber content of a fresh cob is typically between 20 and 30 kg/m3 [17].

This sentence is added in the manuscript: These ratios enable us to maintain the usual proportions of the cob construction, i.e. fiber content of a fresh cob between 20 and 30 kg/m3 [17]

- Are these proportions by volume or by weight “1; 2.5 and 4%)”?? Done

Response: These proportions are by weight

-Lines 152-153, all dimensions should be in millimeter “were carried out on cylinders of Ø11 x 22 cm for the compressive strength test, and on 152 prismatic panels 20 x 20 x 4 cm3 for the thermal tests“ Done

-Line 152, why the authors used these dimensions “Ø11 x 22 cm” which did not agree with any standard

Response: These are the usually used dimensions in the earth-construction scientific community. The mechanical measurements are conducted according to EN 1015–11 standard (Methods of test for mortar for masonry - Part 11 : determination of flexural and compressive strength of hardened mortar, NF EN 1015-11, 2019).

The standard is added in the manuscript.

-Lines 154-155, all dimensions should be in millimeter “on cubic samples 3 x 3 x 3 cm3 for the sorption/desorption test, and Ø11 x 154 22 x 3 cm3 for the water vapor permeability test” Done

-Lines 154-155, why the authors used these dimensions “3 x 3 x 3 cm3 for the sorption/desorption test, and Ø11 x 154 22 x 3 cm3” which did not agree with any standard

Response: The vapor permeability tests are conducted according to the standard ISO 12572 and the sorption desorption according to ISO 12571 standard.

These standards are added in the manuscript.

-Lines 177-178, all dimensions should be in millimeter Done

-In Section 2.1, all properties of the raw materials should be included (for example, the specific gravity, tensile strength, Young’s modulus for each type of fiber should be included)

Response: Table 3 is added in the manuscript with the physical properties of each fiber.

-The abbreviations included in Table 3 should be identified Done

-Lines 269-273, all dimensions should be in millimeter Done

 -In Table 6, please explain why you selected silty soil/sand ratio = 2 and did not use 2.75

Response: The ideal soil for this kind of building consists in a good mix of different sizes of granularity, from clay (about 10-40%) to silt (about 10-40%) to sand (about 35-65%) and even very fine gravel. This is the reason why we select this ratio.

-The information listed in Table 6 are vague

Response: we replaced SF by seaweed fiber and FF by flax fiber

-In Table 6, please explain the unit for this Table. For example, in R0, the silty soil is 21.84 g or kg or weight ratio or volume ratio….??/ Done

-It is better to include Table 6 in kg/m3 

Response: for mor clarity, compositions are reported in term of contained elements weights

-In Table 6, the ratio of fibers by weight or by volume (by weight) Done

-Please check the title of Y-axis of Fig. 6 Done

-Please include the error bars for Figs 3, 5, 7, 8 Done

-Please check page 17

Response: Page 19 (17 in the submitted manuscript) has been checked and revised and Table 7 adjusted.

-Conclusions should be condensed

Response: As requested by the reviewer, the conclusion is condensed

Reviewer 2 Report

The manuscript entitled „Earth Based Building Incorporating Sargassum Muticum Seaweed: Mechanical and Hygrothermal Performances” is a paper of good quality. It is very interesting and deals with an important and recent problem of sustainable building materials modifications using natural raw materials. It requires some amendments. I recommend it to be published after a minor revision.

Specified comments:

1. The introduction section is very good. The background is presented and supported with recent research. Number of references is satisfying. 

2. Materials are sufficiently described. One question rises: What abuou chlorides? Since algae are originated from the sea, the chlorides should be present. Was algae washed, or treated somehow to remove chlorides? It is not mentioned in the article

3. I think that for the clarity of presentation soil 2 should be included in table 4. – for authors decission 

4. Table 6 – no unit specified

5. Unfortunately figure 3 is cut probably by pdf conversion, so is not clearly visible. From fragments of R0 and R1 that are visible - for sure measurements are too small.

6. The methodology is good, but there is no information on number of samples used in the research. Were the averages calculated? Can you present at least standard variations for results?

7. The results are presented clearly and the conclusions are properly derived.

Some spelling mistakes are listed below:

Line 37: “methodically [3-8] Individuals” – “.” is missing

Line 77: “unflower” – „sunflower”?

Line 86: “Britany” meaning “Bretagne (fr.)” – so it should be “Brittany” or if reffering to British Isles it should be Britain.

Line 103: m3 – superscript.

Line 181: “1362 Kg/m3.” – small “k”. more of it later on.

Lines 207, 247, 261: unnecessary line breaks

Line 305: “10mm” – space missing

Line 314, 350: „5 °C” – unnecessary space ?

Equation (1): 0, air, dry, oil should be presented as subscripts. Similarly in other equations and also in some places in the text

Figure 6: 6.000 – decimal part is not necessary.

Author Response

We would like to thank the reviewers for their thoughtful comments and efforts towards improving our manuscript. In the following, we address comments specific to each reviewer below.

All changes in manuscript are in red color.

The manuscript entitled „Earth Based Building Incorporating Sargassum Muticum Seaweed: Mechanical and Hygrothermal Performances” is a paper of good quality. It is very interesting and deals with an important and recent problem of sustainable building materials modifications using natural raw materials. It requires some amendments. I recommend it to be published after a minor revision.

Specified comments:

  1. The introduction section is very good. The background is presented and supported with recent research. Number of references is satisfying. 

Response: We would like to thank the reviewer for this comment

  1. Materials are sufficiently described. One question rises: What abuou chlorides? Since algae are originated from the sea, the chlorides should be present. Was algae washed, or treated somehow to remove chlorides? It is not mentioned in the article

Response: These sentences are added in the Materials and methods section:

The sargassum fibers were dipped in the CaCO3 solution (pH 11) for 6 h to eliminate chloride. The fibers are washed with tap water, dried in oven 80°C for 30 min and then dried aired for 1 week.

  1. I think that for the clarity of presentation soil 3 should be included in table 4. – for authors decission 

Response: The properties of Soil 3 are added as suggested by reviewer

  1. Table 6 – no unit specified Done
  2. Unfortunately figure 3 is cut probably by pdf conversion, so is not clearly visible. From fragments of R0 and R1 that are visible - for sure measurements are too small.

Response: This sentence is added in the Mix design section: For each selected composition, four replicates were prepared.

  1. The methodology is good, but there is no information on number of samples used in the research. Were the averages calculated? Can you present at least standard variations for results? Done
  2. The results are presented clearly and the conclusions are properly derived.

Response: We would like to thank the reviewer for this comment

Some spelling mistakes are listed below:

Line 37: “methodically [3-8] Individuals” – “.” is missing Done.

Line 77: “unflower” – „sunflower”? Done

Line 86: “Britany” meaning “Bretagne (fr.)” – so it should be “Brittany” or if reffering to British Isles it should be Britain. Done

Line 103: m3 – superscript. Done

Line 181: “1362 Kg/m3.” – small “k”. more of it later on. Done

Lines 207, 247, 261: unnecessary line breaks Done.

Line 305: “10mm” – space missing Done.

Line 314, 350: „5 °C” – unnecessary space ? Done.

Equation (1): 0, air, dry, oil should be presented as subscripts. Similarly in other equations and also in some places in the text Done.

Figure 6: 6.000 – decimal part is not necessary. Done.

Reviewer 3 Report

The work is essentially experimental-based. It would be interesting to have a theory behind the results, to confirm that the findings are reasonable. The authors report that numerical simulation results have been found, however they are not presented in this paper, while it should be useful to see them. 

Author Response

We would like to thank the reviewers for their thoughtful comments and efforts towards improving our manuscript. In the following, we address comments specific to each reviewer below.

All changes in manuscript are in red color.

The work is essentially experimental-based. It would be interesting to have a theory behind the results, to confirm that the findings are reasonable. The authors report that numerical simulation results have been found, however they are not presented in this paper, while it should be useful to see them. 

Response : The results of the numerical simulation are reported on Figure 10. This latter represents a comparison between the heating and cooling needs in a housing designed with an external envelope made with algae-based cob and a flax-based cob. This Dynamic Thermal Simulation was performed on Pleiades software. The idea behind this simulation is to perform a rough estimation of the energy savings that algae can allow when considered in cob construction.

Reviewer 4 Report

This study investigate the impact of incorporating sargassum mitucum seaweed fiber in replacement of flax fiber used for standard structural cob. Thus, cob specimens were elaborated and analyzed to evaluate their compressive and hygrothermal performances.

The article is well research and contains novel idea that adds some information to the body of knowledge. Likewise, the paper complies with the writing standard of the Journal and all tests were done according to the normal standard of tests. Based on these aforementioned, I recommend that the research paper can be accepted for publication after the minor revisions is corrected according to suggestions.

1. Please use the new Roman font for the full text, including punctuation marks.

2. When multiple words are juxtaposed, add commas before and, such as lines 222 and 241.

3. Please add a period in front of 267 lines of cm.

4. There is a significant separation between Figure 4 and the caption, please supplement it again for review.

5. The layout of the author's paper has significant problems, especially in lines 590 and 630. Please adjust it.

6. The author conducted an energy assessment in Section 3.3. How did the author obtain these relevant data and whether they were accurate? The author needs to explain in detail in the article.

7. The author's conclusions are too complex. Please refine your conclusions further.

Author Response

We would like to thank the reviewers for their thoughtful comments and efforts towards improving our manuscript. In the following, we address comments specific to each reviewer below.

All changes in manuscript are in red color.

  1. Please use the new Roman font for the full text, including punctuation marks.

Response: We used the Palatino Linotype front as requested by the journal (template of Buildings journal)

  1. When multiple words are juxtaposed, add commas before and, such as lines 222 and 241.

Response: Commas are added before all ‘and’ in the manuscript when multiple words are juxtaposed

  1. Please add a period in front of 267 lines of cm.

Response: Done. See lines 292-293

  1. There is a significant separation between Figure 4 and the caption, please supplement it again for review.

Response: Done. Figure 6 (Figure 4 in the submitted version) has been checked and revised.

  1. The layout of the author's paper has significant problems, especially in lines 590 and 630. Please adjust it.

Response: Page 19 has been checked and revised and Table 7 adjusted.

  1. The author conducted an energy assessment in Section 3.3. How did the author obtain these relevant data and whether they were accurate? The author needs to explain in detail in the article.

Response: These data were obtained by performing a numerical simulation on Pleiades software in order to evaluate the eventual effect of algae on energy needs of a cob building. The estimation of needs is based on the Pleiades calculation code. This code has already been tested and validated by specialized organizations and is commonly used by industries for thermal studies.

Hereafter the sentences added into the “Energy needs” section of the manuscript:

“The characteristics of the walls were derived from experimental characterization data.”

“Also, a representative occupation scenario was defined. In this sense, it was supposed that the housing is unoccupied between 10:00 am and 06:00 pm for the worked days except for Wednesday where the inoccupation is supposed between 10:00 am and 2:00 pm. For the weekends, the housing was supposed occupied all the day. Furthermore, two levels of occupation were imposed: sleeping time (0.022 occup./m2)  and living time (0.016 occup./m2).”

  1. The author's conclusions are too complex. Please refine your conclusions further.

Response: As requested by the reviewer, the conclusion is condensed

Round 2

Reviewer 1 Report

Most of the previous comments have been addressed

Reviewer 3 Report

The paper has been improved after reviewer observation. It can be accepted for publication